# Exploring Short and Efficient Synthetic Routes Using Titanocene(III)-Catalyzed Reactions: Total Synthesis of Natural Meroterpenes with Trisubstituted Unsaturations

**DOI:** 10.3390/molecules27082400

**Published:** 2022-04-08

**Authors:** Jennifer Rosales, Gustavo Cabrera, José Justicia

**Affiliations:** 1Department of Organic Chemistry, Faculty of Sciences, University of Granada, C.U. Fuentenueva s/n, 18071 Granada, Spain; jcrosales@correo.ugr.es; 2Centro de Química Orgánica, Escuela de Química, Facultad de Ciencias, Universidad Central de Venezuela, Caracas 1058, Venezuela; gustavo.g62@gmail.com

**Keywords:** terpenes, natural product, titanocene, synthetic methods, radicals

## Abstract

The stereo- and regioselective total syntheses of OMe derivatives of the scarce bioactive meroterpenoids makassaric acid (**1**) and fascioquinol B (**2**) have been accomplished. The synthetic sequences are based on the following three efficient and selective catalytic reactions: Cu-catalyzed addition of Grignard compounds to an epoxide; a regioselective Barbier-type reaction, catalyzed by Cp_2_TiCl; and regio- and stereoselective bioinspired cyclization, also catalyzed by Cp_2_TiCl. These three key processes allow us to obtain the main skeletons of **1** and **2** in a few steps. The valuable synthetic proposal shown in this work provides fast access to scarce, structurally complex meroterpenes with promising biological activities, which are a sustainable source for later studies and applications in medicine.

## 1. Introduction

The use of organic natural compounds as a source of drugs is tremendously widespread in our society; therefore, these types of compounds are an inexhaustible source of molecules that contribute to improving the living conditions in our society [1]. In this sense, except in those cases in which it has been possible to develop artificial cultivation processes or suitable biotechnological techniques for mass production, there is a very general problem in relation to these compounds—their lack of availability in large quantities. This fact constitutes a critical limitation for the extensive study of promising compounds and their subsequent application in medicine. On the other hand, the total synthesis of natural products with complex structures and applications of interest is one of the main motivations of organic chemistry [2]. The advances in this field have allowed us to achieve synthetic goals that were unthinkable in the past, thanks, in turn, to the development of new reactions that function as useful tools for complex synthesis. Therefore, the combination of both factors is crucial in the advancement of organic chemistry, for achieving increasingly complex synthetic objectives, and, of course, for developing new products (drugs, materials, etc.). However, the preparation of drugs based on organic natural compounds, using organic synthesis, needs to be highly efficient, selective, and as short as possible to have industrial applications. These requirements force the development of new synthetic routes, which allow us to obtain the target natural compounds quickly and efficiently. In this sense, reactions promoted by titanocene(III) complexes have been revealed as a powerful tool in organic synthesis, generating new methods of CC bond formation that proceed through free radical chemistry at room temperature, under mild conditions, and are compatible with numerous functional groups [3,4,5]. Titanocene(III) complexes are capable of generating radicals on carbon atoms from various substrates, such as allyl halides and carbonates [6], α, β-unsaturated carbonyl compounds [7], ketones and their derivatives [8], and, especially, epoxides [4,5]. In this context, bioinspired radical cyclizations from epoxides, catalyzed by Cp_2_TiCl, are an important tool in organic synthesis. This reaction has allowed the preparation of several natural terpenoids in a few steps [4,5]. In particular, it has introduced the possibility of preparing polycyclic compounds, with a specific double bond formed in the final step of cyclization, unselectively accessible by cationic-type processes [9,10,11], which normally lead to the formation of mixtures of regioisomers. Although this synthetic strategy has been widely applied in the preparation of terpenes with an exocyclic double bond [5], the regioselective protocol for the synthesis of compounds with endocyclic unsaturations has been underused until now [12,13]. The application of this modification of Cp_2_TiCl-mediated bioinspired cyclizations would allow us to open short routes to access non-trivial structures.

In this context, we consider that natural bioactive meroterpenes are interesting objectives for this application, such as makassaric acid (**1**) and fascioquinol B (**2**) (see Figure 1), which both have an endocyclic double bond in the C ring. Makassaric acid (**1**) is a scarce meroditerpenoid (only 4.3 mg of **1** for 79 g sponge) that was first isolated from extracts of Indonesian *Acanthodendrilla* sp., an inaccessible sponge located at a depth of 10–15 m [14]. Interestingly, this compound showed good activity as an inhibitor of MAPKAP kinase 2 (MK2), an enzyme involved in the regulation of TNF-α [15], which is a cytokine involved in inflammatory processes. For that reason, makassaric acid (**1**) is claimed to be a potential drug for treating inflammatory diseases, such as rheumatoid arthritis [14]. On the other hand, fascioquinol B (**2**) is also a scarce metabolite isolated from the Australian deep-water (around 100 m) marine sponge *Fasciospongia* sp. [16]. Compound **2** also has a meroditerpenoid skeleton, and it is believed that it is a derivative from the hydrolysis of natural fascioquinol A. This compound also presented interesting biological activities, displaying promising Gram-positive selective antibacterial properties toward *Staphylococcus aureus* (IC_50_ 0.95 μM and *Bacillus subtilis* (IC_50_ 0.30 μM) [16].

In the literature, only the semi-synthesis of **1** from natural terpene sclareol is described, carried out by Basabe, Urones et al. [17]. In this semi-synthesis, the authors obtained **1** following a synthetic sequence that involved 13 steps from sclareol to an intermediate with the terpene skeleton present in **1**. The manipulation of the aromatic fragment of the meroterpene increased the number of steps to 20 [17]. On the other hand, no synthesis of fascioquinol B has been described to date.

The promising biological activities described for **1** and **2**, the fact that they are scarce and elusive compounds, and the challenge of developing short and efficient synthetic sequences based on titanocene(III)-catalyzed reactions, which would allow access to the terpene skeleton of these compounds, are interesting synthetic objectives. Thus, in this article, we describe the preparation of OMe derivatives of compounds **1** and **2**, which include all the structural motifs that exist in the terpene skeletons and in the aromatic moieties, opening new synthetic routes applicable to such compounds.

## 2. Results and Discussion

### 2.1. Retrosynthetic Analysis

Meroterpene-type compounds are characterized by a structure with two perfectly differentiated parts. On the one hand, the terpenoid fragment, which includes various rings of different sizes (normally six-membered rings), while, on the other hand, this type of compound has an aromatic fragment. This is the reason why this type of natural product is considered to have mixed biogenesis (polyketide–terpenoid). This clear structural differentiation has allowed the development of at least two main strategies to address the synthesis of polycyclic meroterpenes. Thus, numerous syntheses based on a two-synthon strategy have been described, such as that introduced by Corey in his synthesis of K-76 [18]. This approach has been extensively used in the preparation of meroterpenes, despite the complex procedures, often with numerous steps, necessary for generating terpenic synthons. In contrast, there is a second, more direct possibility, which consists of the bioinspired cyclization of prenylated aromatic moieties, originally used by González et al. in the synthesis of dl-taondiol [19]. This strategy is shown to be more efficient and straightforward, fulfilling the requirements of selectivity and step economy necessary for developing optimal synthetic processes [20,21]. Based on these considerations, we decided to accomplish the preparation of two closely related derivatives of **1** and **2**, such as compounds **3** and **4**, using a bioinspired cyclization-based protocol, which would allow us to open a new pathway to access this specific group of meroterpenoids, characterized by endocyclic unsaturation at C12.

The retrosynthetic proposal for the preparation of **3** and **4** is depicted in Figure 1.

Compounds **3** and **4**, which are closely related to **1** and **2**, respectively, were constructed from cyclization products **5** and **6**, by removing the hydroxyl group at C3 using the Barton–McCombie protocol [22], and protective group cleavage. Compounds **5** and **6**, which can be considered key products in this synthesis, were prepared from epoxypolyenes **7** and **8**, respectively, using bioinspired radical cyclizations catalyzed by Cp_2_TiCl, which has been widely studied and developed by our research group [4,5]. This reaction allowed access to complex terpenic skeletons with complete regio- and stereoselectivity. It is important to note that only an endocyclic alkene at C12–C13 was obtained in the last step of cyclization, as we previously described [12]. On the other hand, epoxypolyenes **7** and **8** were synthesized following a sequence of four steps, from aldehydes **11**–**12** and commercially available *E*,*E*-farnesyl chloride(**13**), using a titanocene(III)-catalyzed Barbier-type reaction [6], acetylation of the hydroxyl group, epoxidation under basic media [23], and, later, acetylation. Finally, aldehydes **11**–**12** were obtained from aromatic bromides **14** [24] and **15**, and from isoprene monoxide (**16**), using a highly stereoselective Cu-catalyzed addition of corresponding Grignard compounds to epoxide **16** [25], followed by oxidation of the generated allylic alcohols. The proposed retrosynthetic analysis suggests that **3** and **4** might be obtained straightforwardly, in a few steps, compared with the previously described proposals, as expected in bioinspired cyclization-based approximations, opening a short and efficient pathway for such compounds.

### 2.2. Synthesis of OMe Derivative of Makassaric Acid (**3**)

The synthesis of the OMe derivative of makassaric acid **3** was accomplished following the strategy shown in Figure 2. This synthetic procedure presented the following two well-defined parts: (**a**) the synthesis of epoxypolyene **7**, which included all the required structural motifs; and (**b**) the preparation of tricyclic meroterpene **5**, a key intermediate in the proposed sequence, and its subsequent transformation into **3**.

The synthesis of compound **7** began with the addition of the Grignard derivative of **14** (generated using Turbo Grignard [26] in THF) to commercially available isoprene monoxide (**16**), catalyzed by CuBr·DMS, following the described procedure [25]. This reaction allowed us to obtain *E*-allylic alcohol **17** with a high yield (92%) and near-complete stereoselectivity (up to 95% *E* isomer). The oxidation of **17** under soft conditions, using Dess–Martin periodinane (DMP) [27], yielded aldehyde **11**, which was used in the preparation of the key epoxypolyene **7**. Thus, the Barbier-type reaction, catalyzed by Cp_2_TiCl [6], of **11** with commercially available *E*,*E*-farnesyl chloride (**13**) yielded polyene **9** with a moderate yield (51%). This catalytic reaction was especially important in accomplishing this strategy, because it allowed the synthesis of complex polyenes with different substitution patterns, including aromatic rings. Additionally, the generated hydroxyl group was located at C12 (based on terpenes numeration, see Figure 1), which was essential for the latter’s generation of the required endocyclic alkene [12]. Once the main skeleton of the polyene was obtained, we finished the synthesis of the key intermediate **7** using van Tamalen’s protocol [23] for regioselective epoxidation (and also deacetylation at C12), and we subsequently conducted acetylation of the hydroxyl group. Once epoxide **7** was in our hands, we submitted it to bioinspired radical cyclization, catalyzed by titanocene(III), which has been extensively studied and applied by us in natural terpene synthesis [5,12]. Under these soft reaction conditions, we successfully obtained tricyclic meroterpene **5** with a 43% yield. This cyclization occurred with complete stereo- and regioselectivity, and generated three *trans*-fused six-membered rings and six stereocentres in only one step. In this context, it is important to note the controlled formation of the endocyclic double bond at C12–C13 in the last step of cyclization. As we previously described, this alkene formation was due to Cp_2_TiCl-mediated radical fragmentation of β-acetoxy radicals (Figure 3 and (i)) [12,28]. This fact is complementary to the formation of exocyclic alkenes that have also been obtained in several examples of bioinspired radical cyclizations, catalyzed by Cp_2_TiCl [4,5], and allows the synthesis of a wide group of natural terpenes with this functionalization. Additionally, this cyclization has two more important characteristics, which showed its high usefulness. First, the use of smooth reaction conditions and radical-based intermediates is completely compatible with the presence of the essential OAc group in the starting polyene, in opposition to cationic cyclizations [9,10,11], which do not allow the introduction of oxygenated functions in these compounds. Second, tertiary carbocations generated in typical cationic cyclizations are electrophilic enough to attack positions of the aromatic ring [29,30], leading to undesired byproducts. However, radical cyclizations, catalyzed by Cp_2_TiCl, occur without changes in the aromatic synthon, and are not in competition with the desired termination step [31,32,33] (see Figure 3). This fact represents a significant difference, not only concerning carbocationic processes, but also in contrast with radical cyclizations promoted by other transition metals or PET processes [34].

With all this information in mind, we can conclude that bioinspired radical cyclization, catalyzed by Cp_2_TiCl, shows, again, its great potential in the field of natural terpene synthesis, offering solutions to unresolved problems in other kinds of cyclization processes and allowing us straightforward access to complex terpenic skeletons.

Once the cyclization compound **5** was obtained, the synthesis carried on removing the hydroxyl group at C3 through the Barton–McCombie protocol to yield **19** [22], –OTBDMS cleavage using *n*Bu_4_NF, and the further oxidation of the generated primary alcohol with DMP and Pinnick conditions [35]. Thus, we obtained the OMe derivative of makassaric acid **3** in only 12 steps and with a 5% global yield. A comparison with the previously described semi-synthesis of **1** indicated that our route decreased the synthetic sequence in eight steps, until an analog intermediate was obtained. This result confirmed that our initial aim, designing and executing a more efficient and straightforward synthesis of meroterpenes, has been reached in the case of compound **3**.

### 2.3. Synthesis of OMe Derivative of Fascioquinol B (**4**)

As we described in Section 2.2, the application of three key, highly stereo- and regioselective catalytic procedures has allowed us to open efficient access to meroterpenoid-type compounds, with **3** being a good example. However, the application of this proposed route to the synthesis of other examples of meroterpenoids, to confirm the generality of the process, would be desirable. Thus, here, we also propose the preparation of other examples of meroterpenes, such as compound **4**, a close derivative of fascioquinol B (**2**). The synthesis of **4** is depicted in Figure 4.

The synthesis started with the previously mentioned prenylation of aromatic **15**, via the addition of the corresponding Grignard derivative to isoprene monoxide **16** [25]. Thus, the corresponding allylic alcohol **20** was obtained with a nearly quantitative yield and excellent stereoselectivity (up to 95% *E* isomer). Oxidation with DMP [27] generated aldehyde **12**, which was used in the construction of the required hydroxylated polyene. For this, compound **12** was submitted to a Barbier-type reaction, catalyzed by Cp_2_TiCl [6], to yield alcohol **10** (see Figure 1) with a good yield (60%). Subsequently, the protection of the hydroxyl group at C12, as acetate, generated the desired polyene **21**. Once compound **21** was in our hands, we accomplished the synthesis of the required epoxide for later cyclization, following the classical regioselective procedure described by van Tamelen [23], and with further protection of the hydroxyl group as acetate, to yield compound **8** with a 57% yield after two steps. Bioinspired radical cyclization, catalyzed by titanocene(III) [12], generated the corresponding tricyclic compound **6** in a completely regio- and stereoselective reaction. The yield of this reaction (35%) can be considered acceptable, considering the number of CC bonds and stereocentres formed in only one step. All the considerations indicated for the cyclization of **7** to yield **5** (Figure 2 and Figure 3), as the selective formation of endocyclic unsaturation, can also be applied to this reaction. Finally, the synthesis of compound **4** was completed using the two-step Barton–McCombie protocol [22], which removed the hydroxyl group located at C3 (52% yield and two steps). Thus, we could obtain meroterpene **4**, closely related to **2**, in only nine steps and with a 5.8% global yield. The synthesis of 2 would be completed using the previously described oxidation-reduction sequence [36].

## 3. Experimental Section

### 3.1. General Details

Deoxygenated solvents and reagents were used for all reactions involving Cp_2_TiCl. THF and benzene were freshly distilled from Na. Other dry solvents, such as CH_2_Cl_2_ and 1, 2-dichloroethane (DCE), were acquired from commercial supplier (Sigma-Aldrich, Steinheim, Germany) (Cp_2_TiCl_2_ was acquired from a commercial supplier (Alfa Aesar, A11456, Thermo Fisher Scientific, Kandel, Germany). Products were purified by flash chromatography on VWR silica gel (40–60 μm). Yields refer to analytically pure samples. NMR spectra were recorded in NMR Varian Direct Drive (400 MHz or 500 MHz, Agilent Technologies, Santa Clara, CA, USA) spectrometers. The following known compounds were isolated as pure samples and showed NMR spectra that matched those of the reported compounds: **12** [37], **14** [24], and **20** [38].

### 3.2. Main General Procedures

#### 3.2.1. General Procedure for Cu-Catalyzed Grignard Compound Additions (GP-1) 

The corresponding Grignard derivative (1.2 mmol, previously prepared by treatment of the aryl bromide **14**–**15** (1 mmol) with Mg (1.4 mmol) in dry THF (3 mL) at 50 °C, or with iPrMgCl·LiCl (1.3 mmol) in dry THF (3 mL) at 0 °C) was added over 3 h via a syringe pump to a −42 °C solution of **16** (1 mmol), and CuBr·DMS (0.5 mmol) in dry THF (15 mL). The mixture was stirred for an additional 16 h at room temperature. Then, Et_2_O (20 mL) was added and the mixture was washed with a saturated solution of NH_4_Cl, dried (anhyd. Na_2_SO_4_), and the solvent was removed. Products **17** and **20** were isolated by flash chromatography of the residue (mixtures of hexane/EtOAc) and characterized using spectroscopic techniques. See Appendix A for more details.

#### 3.2.2. General Procedure for Barbier-Type Prenylation of Aldehydes Catalyzed by Cp_2_TiCl (GP-2)

Strictly deoxygenated THF (20 mL) was added to a mixture of Cp_2_TiCl_2_ (0.2 mmol) and Mn dust (8 mmol) under an Ar atmosphere, and the suspension was stirred at room temperature until it turned lime green (after about 15 min). Then, 2,4,6-collidine (6 mmol) and Me_3_SiCl (4 mmol) were added. Subsequently, a solution of carbonyl compounds **11** or **12** (1 mmol) and farnesyl chloride (**13**) (2 mmol) in THF (2 mL) was slowly added over a period of 2.5 h, and the solution was stirred for 6 h. The reaction was then quenched with a saturated solution of NaHCO_3_ and extracted with EtOAc. The organic layer was washed with brine, dried (anhyd. Na_2_SO_4_), and the solvent was removed. Products were purified by flash chromatography on silica gel (mixtures of hexane/EtOAc) and characterized using spectroscopic techniques. See Appendix A for more details. In some experiments, trimethylsilyl derivatives were observed. In these cases, the residue was dissolved in THF (20 mL) and stirred with *n*Bu_4_NF (10 mmol) for 2 h. The mixture was then diluted with EtOAc, washed with brine, dried (anhyd. Na_2_SO_4_), and the solvent was removed.

#### 3.2.3. General Procedure for Bioinspired Cp_2_TiCl-Catalyzed Cyclization Reactions (GP-3)

Strictly deoxygenated THF (20 mL) was added to a mixture of Cp_2_TiCl_2_ (0.2 mmol) and Mn dust (8 mmol) under an Ar atmosphere, and the suspension was stirred at room temperature until it turned lime green (after about 15 min). Then, a solution of epoxypolyene **7** or **8** (1 mmol), 2,4,6-collidine (6 mmol) in THF (2 mL), and Me_3_SiCl (4 mmol) was added, and the mixture was stirred for 16 h. The reaction was then quenched with 2 N HCl and extracted with EtOAc. The organic layer was washed with brine, dried (anhyd. Na_2_SO_4_), and the solvent was removed. Products **5** and **6** were isolated by flash chromatography of the residue (mixtures of hexane/EtOAc) and characterized using spectroscopic techniques. See Figure 1 for details.

## 4. Conclusions

We have developed a short and efficient synthetic sequence for the synthesis of biologically active natural meroterpenoids with trisubstituted endocyclic unsaturation. Our strategy relies on three key metal-catalyzed reactions, such as the Cu-catalyzed addition of Grignard derivatives to isoprene monoxide, the Ti-catalyzed Barbier-type reactions of α, β-unsaturated aldehydes with complex farnesyl chloride, and the bioinspired Cp_2_TiCl-catalyzed cyclizations of functionalized epoxypolyenes. The combination of these catalytic procedures, which proceed under mild reaction conditions and with high regio- and stereoselectivity, allowed us access to the meroterpenoid skeletons in short synthetic sequences. The synthetic approximation could be considered an efficient and short route for the preparation of this specific class of meroterpenoids, in comparison with other existing protocols, offering an inexhaustible source of these compounds for their further application in biological and pharmacological studies.

## Data Availability

Data is contained within the article or Appendix A.

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
