# Peer review of "Exploring Short and Efficient Synthetic Routes Using Titanocene(III)-Catalyzed Reactions: Total Synthesis of Natural Meroterpenes with Trisubstituted Unsaturations"

_molecules, 2022, doi:10.3390/molecules27082400_

Round 1
Reviewer 1 Report
This work describes the synthesis of O-methyl derivatives of the natural products makassaric acid and fascioquinol B with a titanium(III)-catalyzed tandem terpene cyclization as key step. The same authors have previously applied this approach to several other terpene natural products. This work again underlines the significance for this elegant cyclization reaction in steroid and related molecule synthesis. The results are sound and the supporting information contains all relevant procedures and characterization data. The manuscript itself is well-written, but it could be more concise (see remarks below).
Overall, I recommend publication of this work in molecules, because it showcases the importance of this type of low-valent titanium-catalyzed reactions in organic synthesis. The step size is significantly reduced if compared to previous syntheses of the target compounds. I have only a few remarks that could be addressed in form of a minor revision before the publication of this manuscript.
1) The introduction (in particular lines 23–44) seems rather long and general. The authors may wish to shorten or delete this part, also the connection to the current COVID-19 situation seems a bit artificial.
2) Did the authors attempt to complete the total syntheses of compounds 1 and 2? Was the removal of the O-methyl groups attempted? The authors could comment on that point. If it did not work, it would be good to know, why. The preparation of the natural products would also have allowed a comparison of the spectra/the data of the synthetic material with the original data.
3) The contents of chapter 2.3 are quite related to chapter 2.2, it is basically the same route again with a different precursor and only minor modifications. As a suggestion, both syntheses could be combined in one Scheme for reasons of conciseness.
4) In the synthesis of 20 from 15, why was magnesium used and not turbo-Grignard as before for 14-->17?
5) page 4, Scheme II and text (page 5, line 162): One could note in the text that the K2CO3/MeOH treatment after the bromination of 18 with NBS led to a deacetylation and required the reinstallation of the acetyl group.
6) page 4, line 132. Suggestion: “Barbier type reaction”
7) page 5, line 161 and page 6, line 218: “van Tamalen” (v not capitalized?)
8) page 5, line 174-175: The authors write that “the cyclization is completely compatible with the presence of the essential -OAc group” — If I am not mistaken, the exact opposite is the case: The acetoxy group is removed in the process. I suggest rewriting this passage to avoid confusion.
9) page 5, line 190. Suggestion: “Once the cyclization product 5 was obtained, ...”
10) page 4, line 160. Suggestion: “Once the main skeleton of the polyene was obtained,...”
11) Supporting Information: Please scale all NMR spectra in such a way that the largest compound signal spans the whole Y-axis of the spectrum (not the solvent signal). The 1H NMR spectra should cover a range of at least –0.5 - 10.5 ppm. The 13C NMR should cover at least –20 - 220 ppm.
Author Response
Criticism 1. “The introduction (in particular lines 23–44) seems rather long and general. The authors may wish to shorten or delete this part, also the connection to the current COVID-19 situation seems a bit artificial.”
Reply to criticism 1. Following the indications of Referee 1, the Introduction section has been shortened.
Criticism 2. “Did the authors attempt to complete the total syntheses of compounds 1 and 2? Was the removal of the O-methyl groups attempted? The authors could comment on that point. If it did not work, it would be good to know, why. The preparation of the natural products would have also allowed a comparison of the spectra/the data of the synthetic material with the original data.”
Reply to criticism 2. The main objective of this work is to develop an efficient route to the synthesis of meroterpenes with trisubstituted alkene in the last ring generated in the cyclization (C-ring in this case). The synthesis of O-methyl groups derivatives 3 and 4 are examples of the proposed routes. However, mainly in the case of fascioquinol B, we also were interested on to complete the synthesis of the natural product, to accomplish the first total synthesis of this compound. With 4 in our hand, we tried removing O-methyl groups using a known one-step methodology, described for a related compound in Nat. Chem. 2020, 12, 173–179. Unexpectedly, the reaction conducted on the destruction of 4.
Criticism 3. “The contents of chapter 2.3 are quite related to chapter 2.2, it is basically the same route again with a different precursor and only minor modifications. As a suggestion, both syntheses could be combined in one Scheme for reasons of conciseness.”
Reply to criticism 3. Synthetic routes in chapter 2.2 and 2.3 are very similar. However, the first and the last steps in the synthesis of 3 and 4 are different. In our opinion, combinations of both synthetic routes in only one scheme could be confused. If referee 1 accepts, we proposed to keep Schemes II and IV to show all the specifications of both routes.
Criticism 4. “In the synthesis of 20 from 15, why was magnesium used and not turbo-Grignard as before for 14-->17?”
Reply to criticism 4. The use of the described reaction conditions for the synthesis of Grignard derivatives of 14 and 15 was due to the different reactivity of both aromatic compounds. Thus, the reaction of 15 with turbo-Grignard was completely unsuccessful, and we must use Mg and heating. However, the reaction of 14 with Mg was also unsuccessful under several conditions, yielding the best results when turbo-Grignard was employed.
Criticism 5. “page 4, Scheme II and text (page 5, line 162): One could note in the text that the K2CO3/MeOH treatment after the bromination of 18 with NBS led to a deacetylation and required the reinstallation of the acetyl group.”
Reply to criticism 5. Exactly. We have included a comment on this fact in line 151 (new version of the manuscript).
Criticism 6. “page 4, line 132. Suggestion: “Barbier type reaction””
Reply to criticism 6. The proposed corrections have been made.
Criticism 7. “page 5, line 161 and page 6, line 218: “van Tamalen” (v not capitalized?)”
Reply to criticism 7. The proposed correction has been made.
Criticism 8. “page 5, line 174-175: The authors write that “the cyclization is completely compatible with the presence of the essential -OAc group” — If I am not mistaken, the exact opposite is the case: The acetoxy group is removed in the process. I suggest rewriting this passage to avoid confusion.”
Reply to criticism 8. The comment indicated on page 5, line 174-175, is referred to the fact that the presence of the OAc- group in the polyene is allowed when radical reactions conditions are used. Of course, OAc- group is removed in the termination step, but not other collateral reactions are observed. However, under acidic reaction conditions for cation generation, allylic oxygenated functions in the starting polyene are usually incompatible (Ref. [9-11]). Nevertheless, the text has been modified to avoid confusion.
Criticism 9. “page 5, line 190. Suggestion: “Once the cyclization product 5 was obtained, ...””
Reply to criticism 9. The proposed corrections have been made.
Criticism 10. “page 4, line 160. Suggestion: “Once the main skeleton of the polyene was obtained,...””
Reply to criticism 10. The proposed corrections have been made.
Criticism 11. “Supporting Information: Please scale all NMR spectra in such a way that the largest compound signal spans the whole Y-axis of the spectrum (not the solvent signal). The 1H NMR spectra should cover a range of at least –0.5 - 10.5 ppm. The 13C NMR should cover at least –20 - 220 ppm.”
Reply to criticism 11. We have changed all NMR spectra in SI applying the comments of Referee 1.

Reviewer 2 Report
- Acronis such as TBDMS such be explained
- The source of Cp2TiCl2 should be specified
- IR spectrums for all synthetised compounds should be registered, next to NMR and MS analysis
- The application potential of the proposed synthetic protocol should be described in the conclusion, within the background of the alternatively ways.
Author Response
Criticism 1. “Acronis such as TBDMS such be explained”
Reply to criticism 1. Acronym TBDMS means “t-butyldimethylsilyl”. The explanation has been added in Scheme I.
Criticism 2. “The source of Cp2TiCl2 should be specified”
Reply to criticism 2. Cp2TiCl2 was acquired from commercial supplier (Alfa Aesar, A11456). We have included this information in 3.1 General Details.
Criticism 3. “IR spectrums for all synthetised compounds should be registered, next to NMR and MS analysis”
Reply to criticism 3. For the characterization of new synthesized compounds, we have followed the “Instructions for Authors” of Molecules. In the section Correct Identification and Characterization of Chemical Compounds, in Organic Compounds, the instruction said: “Reports on previously undescribed organic compounds should include, as supplementary data, 1H, 13C and/or other key heteronuclear or 2D NMR spectra, together with High Resolution Mass Spectrometry (HRMS) or elemental analysis.
For that reason, we only acquired the corresponding 1H, 13C (with DEPT) and HRMS spectra for all new compounds described in the manuscript. IR spectrums were not measured, so we cannot include in the manuscript.
Criticism 4. “The application potential of the proposed synthetic protocol should be described in the conclusion, within the background of the alternatively ways.”
Reply to criticism 4. In Conclusions, we indicated that our synthetic proposal would be useful for accessing to the meroterpenoid skeletons in short synthetic sequences. This is the main application for it. Additionally, we have improved “Conclusions” section in the manuscript including some details about the proposed routes, as they are shorter than other strategies used before.
